# Patients' views about screening for atrial fibrillation (AF): a qualitative study in primary care

Mark Lown  , Christopher R Wilcox, Stephanie Hughes, Miriam Santer, George Lewith, Michael Moore, Paul Little

GL since deceased

Primary Care and Population Sciences, University of Southampton, Southampton, UK

**Correspondence to**
Dr Mark Lown;
M.Lown@soton.ac.uk

## ABSTRACT

**Objectives** There has been increased interest in screening for atrial fibrillation (AF) with commissioned pilot schemes, ongoing large clinical trials and the emergence of inexpensive consumer single-lead ECG devices that can be used to detect AF. This qualitative study aimed to explore patients' views and understanding of AF and AF screening to determine acceptability and inform future recommendations.

**Setting** A single primary care practice in Hampshire, UK.

**Participants** 15 participants (11 female) were interviewed from primary care who had taken part in an AF screening trial. A semistructured interview guide was used flexibly to enable the interviewer to explore any relevant topics raised by the participants. Interviews were recorded, transcribed verbatim and analysed using inductive thematic analysis.

**Results** Participants generally had an incomplete understanding of AF and conflated it with other heart problems or with raised blood pressure. With regards to potential drawbacks from screening, some participants considered anxiety and the cost of implementation, but none acknowledged potential harms associated with screening such as side effects of anticoagulation treatment or the risk of further investigations. The screening was generally well accepted, and participants were generally in favour of engaging with prolonged screening.

**Conclusions** Our study highlights that there may be poor understanding (of both the nature of AF and potential negatives of screening) among patients who have been screened for AF. Further work is required to determine if resources including decision aids can address this important knowledge gap and improve clinical informed consent for AF screening.

**Trial registration number** ISRCTN 17495003.

## INTRODUCTION

Atrial fibrillation (AF) is a common arrhythmia affecting around 10% of people aged over 65 in the UK[1] and is associated with an increased risk of stroke which is substantially reduced by anticoagulation.[2] Across England, it is estimated that 425 000 people are living with undiagnosed AF,[3] and there has been much recent debate about screening for AF.[4] The European Cardiac Society currently recommends opportunistic screening in patients aged >65 and consideration of

### Strengths and limitations of this study

► A strength of this study is that all the interviews were conducted by a single researcher, thus ensuring consistency.
► The study had good representation in terms of a typical screening population aged over 65 years although the high proportion of female participants may have affected the results.
► All of the participants had taken part in the Screening for Atrial Fibrillation using Economical and Accurate TechnologY trial and those not wishing to be screened for atrial fibrillation (AF) may have had different views.
► Participants were recruited from a single surgery and may have had similar views, not representative of the wider population. However, they were from an area of low deprivation, and as they had taken part in a trial, they were likely to be more health literate than the wider population who may have a poorer understanding of AF, although we did not record individual demographic data.
► The participants all had a negative screen which may have affected their attitudes.

systematic screening in patients aged >75 or in those at higher stroke risk.[5] No country has yet implemented a systematic screening programme although National Health Service (NHS) England has commissioned pilot AF screening schemes in pharmacies and in influenza clinics to 'test the treatments and care models of tomorrow.[6 7]

There are ongoing, large randomised controlled trials to investigate the cost-effectiveness of AF screening and the outcomes are eagerly awaited.[8] We are also witnessing a shift in healthcare with increased consumerisation of medical devices and the emergence of relatively inexpensive consumer single-lead ECG devices and watches that can be used for AF detection.[9 10] Hence, there now exists a great need to ensure patients are well informed before undertaking any form of AF screening and to understand patient attitudes

towards screening for AF. In this article, we report on an interview study that aimed to explore patient views on screening for AF.

## METHODS

The current qualitative study was nested within the Screening for Atrial Fibrillation using Economical and Accurate TechnologY (SAFETY) study.[11] Four hundred eighteen participants were recruited to the SAFETY trial from three primary care practices in the Wessex area. Individuals aged over 65 years both with and without a coded diagnosis of AF in their medical records were invited by their general practitioner (GP) to a single nurse-led screening visit to test the accuracy of several devices (a blood pressure meter, a single-lead device and two ECG sensing consumer devices) for the detection of AF. Research nurses explained how to use the devices and participants were able to ask questions about AF and the devices during the visit.

### Data collection

For this qualitative study, a convenience sample of 34 of the 418 trial participants not known to have AF from a single GP surgery were invited to take part in an interview. Participants who consented to the qualitative study were approached sequentially (in order of randomisation to the main study) by telephone (from the first recruiting site of 3 which all had low levels of deprivation). Of these, 15 agreed to participate. We only invited participants who did not have AF as they would likely be more representative of a typical screening population. We did not record reasons for declining to participate. All the participants had previously been sent information leaflets as part of the trial invitation with detailed information on AF, screening for AF and treatment options if AF were detected. Interviews were conducted by SH (a senior researcher with considerable qualitative research experience) via telephone, audiorecorded and transcribed verbatim, assigning identification numbers to preserve anonymity. A semistructured interview guide was used flexibly to enable the interviewer to explore any relevant topics raised by the participants (box 1). The interview guide covered topics such as the patient's understanding of AF, views about AF screening (including benefits and drawbacks), opinions about the devices trialled (eg, device comfort) and opinions about future use of the devices. The interviews were carried out between May 2017 and July 2017 and lasted around 15 min.

### Data analysis

Interview transcripts were analysed using inductive thematic analysis (as we had little prior data or predetermined theory on AF).[12] Transcripts were read and reread (ML, SH, CRW, MS) to identify codes, which were then organised iteratively into a coding manual by ML and CRW. Main themes and subthemes were generated independently by ML and CRW then reviewed and refined

---

| Box 1 | Interview schedule |
| --- | --- |

Can you describe your understanding of atrial fibrillation?
What are your views about screening for atrial fibrillation?
► What do you think the positives of screening are?
► What do you think the negatives of screening are?
What did you think about the study information given to you?
Did you have any reservations about taking part, and if so, can you tell me a bit about them?
Do you have any regrets about taking part? If so, can you tell me a bit about them?
Could you tell me a bit about the devices you tried as part of the SAFETY trial?
How comfortable did you find the devices?
How would you feel about wearing or using the devices for a few weeks for screening?
Is there anything else you would like to share about your participation in the trial or your thoughts about AF?

---

through further discussions within the team, which included a mix of clinicians and academics with varying degrees of experience. The study team held regular meetings during the data collection phase and assessed the data for saturation of main themes and searched for disconfirming cases (the authors reviewed themes emerging during coding and were confident saturation had been reached when no new themes of interest were arising approximately half way through the interviews.) The sample size was appropriate and sufficient to achieve saturation of main themes.[13]

### Patient and public involvement

Patient and public representatives were involved in the design of the study from the funding application stage and in protocol development. All the study materials were developed with lay input.

## RESULTS

### Participant characteristics

Of the 15 participants who took part in the qualitative study, 4 were male (26.7%), and the average age was 68 years (range=65–73) (SD=2.74). The average age of the participants in the SAFETY trial was 73.9 years and 43% were female. The index of multiple deprivation score for the participating GP practice was 17.7 (second quintile in England (from low levels of deprivation to high)) and the income deprivation score affecting older people (>60 years) index was 17.3 (middle quintile). All participants were fluent in English. A number of themes emerged relating to their (1) understanding of AF, (2) attitudes to screening (in general and for AF specifically), (3) attitudes to the screening devices tested during the SAFETY study and (4) their attitudes to undergoing prolonged screening.

### Understanding of atrial fibrillation

Participants were asked to describe their understanding of AF. In interpreting responses to this question, it needs to

---

be highlighted that all participants had received an information sheet, which described AF as 'an irregular heart rhythm that can lead to blood clots forming within the heart which can come loose and cause a stroke'. Despite this, there was considerable confusion about the nature of the condition. Although the majority of participants said they were aware that AF related to a problem with the heart, many (6/15) seemed unaware that it related to heart rhythm irregularity, and few acknowledged its association with risk of stroke or developing clots (2/15).

> Something wrong with your heart system…of the atrium and connecting pipes. (P15)

> Well, to identify a possible stroke and high blood pressure and possible stroke and heart attack. (P5)

> it's a condition that is related to the blood circulation. (P43)

> Well if the heart isn't functioning properly and this is a condition that could be picked up…if you have heart problems. (P65)

### Attitudes to screening

When asked for their views regarding screening for atrial fibrillation, many of the participants stated positive opinions about health screening in general, with regards to early detection and saving money for the health service.

> I believe in screening as much as possible…anything that helps with picking up conditions, I think, is a good thing. (P72)

> Having more treatments or monitoring available… can't be a bad thing. (P122)

> Saves the health service a hell of a lot of money…any kind of health screening on the NHS system is a very good idea. (P131)

Many participants raised positive opinions regarding early detection and treatment of AF specifically, and two mentioned prevention of stroke.

> I think it's sensible to know if you've got AF, because then it's possible to have some treatment. (P39)

> In response to the question 'What are your thoughts about screening for atrial fibrillation': 'Well I actually think it's a good idea; there's a lot of heart conditions in my own family on my father's side, and so it's no harm to be monitored every so often, along the way.' (P53)

> I think it's an excellent idea; better to catch a condition early if you possibly can, especially if people… [are] not aware they have it…if there's medication or other things to stop it…I know there are so many other calls on the Health Service, but I know stroke and so on being so debilitating; if you can avoid any at all, that would be a good thing. (P125)

Most participants did not describe any potential downsides to screening for AF. Possible negatives that were raised by participants included anxiety and cost for the health service.

> It might make you anxious, but my anxiety on that score would be counteracted by—at least I know and it's now in hand, as opposed to not knowing and then it causing a complication. (P39)

> Perhaps…it might trigger people to be too anxious about their health, perhaps if they were …that sort of person. (P125)

> I suppose the cost for the NHS but, in the long-term, if you can pick something up early and correct it, it's going to be cheaper than if it's left and then down the road they are going to need more care and intervention. (P72)

> I suspect it's probably desirable…the cost or the complexity might outweigh the benefits; I'm not completely sure on that. (P122)

Participants may have misunderstood the scope of the screening test. One participant felt reassured they had a healthy heart.

> Well I suppose it was a chance to see if my heart was healthy; also it reassured me. (P65)

### Attitudes to the screening devices

Some participants stated no particular preference towards any device and felt that all were comfortable.

> Absolutely no question of them being unpleasant… they were very unintrusive and unobjectionable. (P122)

> They were all comfortable. I could have coped with any of them…if I'd been selected to use a particular type it wouldn't have bothered me. (P72)

Others participants had mixed opinions regarding the user-friendliness of the devices and stated a preference towards those which were the least uncomfortable, least trouble and least time-consuming to use (although their opinion as to which device was preferred differed between participants).

> I think probably the one on the finger was most convenient to use. There was another one where you had to put patches on yourself…[which] I would probably have to force myself to do at home…it would just be one of those irritating things to do. There was another one…that every time you used it, you were going to have to turn on the computer, log into a site and it would upload the data onto the site… at a specific and regular time…[which would be] a bit of a faff." (P45)

> They were all fine. I think the most difficult one… where you have all the things put on you…that's the one that takes the time. But all the others seemed very, you know, quick…I think the first one was the simplest, the one with the thumbs. (P76)

The worst one was the blood pressure cuff which was quite uncomfortable. The one…you wore it round your [rib cage]—you weren't aware you were wearing it and I think that was the best one…The handheld device was quite comfortable but…you were holding it, so whereas wearing the thing around your chest, your hands were free to do other things…and the ECG, obviously, you had to be fully engaged with that and you couldn't do other things. (P125)

### Attitudes towards undertaking prolonged screening

Many participants stated that they would be happy to undergo prolonged screening using these (or similar) screening devices if it were recommended by their doctor.

I just tend to follow advice with that sort of stuff…if it's because there was a need…it would be in my best interest then I would do yes. (P15)

Even if it was uncomfortable…you would just do what you had to do really…I wouldn't have any hesitation if it was a health matter. (P125)

Others were more reserved about the idea of prolonged screening over a number of weeks or had specific concerns, for instance about the time required and the potential of screening to provoke anxiety or 'take over your life'.

Well I'm not sure. It depends how many weeks we're talking about…I mean I don't have any other particular issue other than the timing. (P131)

I'm probably a bit of an anxious person…so I might not be quite at ease…I'm not the sort of person that happily does something and just forgets…it tends to take over your life a bit…so if [it was] for two weeks or so, I might be a bit hesitant. (P130)

Some participants seemed unclear or doubtful about prolonged screening, that is, they would be happy to test for AF if there was a definitive need identified by their doctor but not just if it was a matter of their age. This suggests that, even if they felt the test was tolerable, they may have reservations about doing this in the absence of symptoms.

I mean if there was a valid reason behind it…if it was just oh you're getting to a certain age and we ought to look at it, I would probably…have a bit of a half-hearted effort at it…but if the doctor…[said that there] might be a problem here and it needs further investigation, then obviously I would take it quite seriously. (P45)

## DISCUSSION
### Principal findings

Participants expressed positive opinions towards screening for AF at an early stage and the potential to save costs although some participants mentioned that screening might provoke anxiety. The participants seemed to have an incomplete understanding of AF (despite receiving printed information sheets about AF, paroxysmal AF and anticoagulation treatment) and conflated it with other heart problems or with raised blood pressure. Although a sphygmomanometer device was employed as one of the screening devices, the other devices used in the trial measured ECG signals, and participants were informed that the 12-lead ECG they had was a definitive diagnostic test. However, more than half of those invited for the screening trial were willing to take part in the SAFETY trial[14] with the information we provided.

Importantly, when asked for their opinions on any potential drawbacks, participants did not mention the accuracy of the diagnostic test or the potential for overdiagnosis. The screening devices were generally well accepted by the participants and found to be unobtrusive, but some participants expressed concerns about comfort, user-friendliness and time taken to use the device.

### Strengths and limitations of this study

A strength of this study is that participants were representative of a typical AF screening population aged over 65 years and had received information about AF and participated in screening. Patients who chose not to participate in the screening trial may however have had different views with respect to the usability and acceptance of the devices or may have chosen not to participate due to concerns over anticoagulation treatment or anxiety regarding the screening process.

A further limitation is that participants were recruited from a single surgery and may therefore hold similar views. However, as they had taken part in a trial of AF screening, were likely to have a better understanding of AF than the general population, making our finding of confusion around the topic particularly interesting.

The high proportion of female participants is a common occurrence in qualitative studies and may have affected the findings so is also a limitation. Furthermore, we were unable to explore the views of screen positive patients who may have felt they could have been better informed prior to screening and may have had different opinions on screening in general.

### Findings in relation to other studies

Previous studies have identified that many patients with AF were not aware of the name of the condition[15] or that it led to an increased risk of stroke.[16] Other work has found that patients were uncertain what AF was before and after outpatient cardiology clinic appointments.[17] They also had difficulty understanding why they were treated with anticoagulation and why treatment was recommended lifelong.[17] Older patients, in particular, may have a poor understanding of AF.[18] This may be a key barrier to accepting anticoagulation treatment and to future treatment adherence, which is imperative for stroke risk reduction.[19] Other work has found many patients were unaware that AF could be asymptomatic and therefore they may not be aware of paroxysmal episodes which could remain undiagnosed.[20]

Other data suggests that screening can induce anxiety in other screening programmes including breast and cervical screening[21 22] and the potential for psychological harm from being labelled with an unexpected diagnosis.[23] However, when directly asked about potential downsides of screening in this study, participants did not mention the potential risk of harm from anticoagulation treatment, or the risk of further investigations when deciding to participate, suggesting that this was not a major concern for them. This is consistent with other studies showing that many patients are unable to identify potential harms from other screening tests, and of those that did, they were mostly related to the test itself not to further testing or treatment.[24] In contrast, patients could name benefits and tended to overestimate them.[23]

Patients do worry about potential false positive results for other screening tests such as lung cancer tests.[25] There is also concern that patients may not understand the concept of overdiagnosis.[20] However, evidence suggests that older patients may be suspicious or resistant to the concept of overdetection of other conditions.[21 26] Interestingly, there is also evidence that older adults perceived overuse to have occurred when interventions were used in the absence of symptoms (excluding cancer screening) did not improve symptoms, or against their preferences.[27]

AF screening devices have been found to be well accepted by participants in previous large-scale trials.[28 29] However, many participants in our study stated they would be happy to undergo prolonged screening only if recommended by their doctor and some participants had specific concerns with respect to the time taken.

### Practice and policy implications

There is a need to provide clear and concise information about AF, and to check patient understanding, before proceeding with screening. Checklists could be used to ensure key points have been discussed and considered, and patients may require time to weigh up the risks and benefits before deciding to proceed with screening. Decision aids have been implemented for AF treatment but have usually been designed to support clinician decisions and do not explicitly engage patients.[30] Decision aids could potentially improve patient knowledge prior to screening and have been used to improve knowledge for other screening programmes.[31–34] They could be used to explain the diagnostic test, conditions that could be diagnosed, quantitative information relating to diagnostic accuracy and the risks and benefits of treatment. They could also be used to promote clarification of patients' preferences about the screening and potential consequences to improve patient knowledge. Healthcare professionals could actively ask about any potential anxiety participants may have prior to screening and be prepared to discuss these.

Although participants in this study did not raise concerns regarding a lack of clear information, the importance of providing information of risks as well as benefits of screening is well established in other screening programmes.[35] For patients considering screening for AF,

healthcare professionals could consider providing information about the risk of bleeding with anticoagulation treatment if AF were detected, in addition to potential benefits. A discussion of available treatment options and lifestyle modifications before undergoing testing might also ensure the potential for future treatment adherence if AF was diagnosed. Patients could also be informed about the specificity of the screening test and reminded that screening will not provide information on general heart function or cardiovascular risk to avoid false reassurance.

### Future research

Further research should focus on screen positive patients to determine their views on information provision prior to screening and their wider opinions on screening for AF. Future research should also include a wider demographic of patients. There is also a need to understand how prescreening information and discussions influence patients' decisions to undergo screening for AF.

## CONCLUSIONS

Our study highlights that, even among patients who have been screened for AF, there may be poor understanding of both the nature of AF and potential negatives of screening. Further work is required to determine if resources including decision aids can address this important knowledge gap and help patients make informed decisions around AF screening.

**Acknowledgements** The authors would like to thank all the participants who took part in this study: without them this study would not have been possible.

**Contributors** ML, GL, MM and PL designed the study. SH conducted the interviews. ML and CRW coded and analysed the interview data. ML, CRW, SH and MS contributed to the interpretation of the data. ML drafted the manuscript, and all authors contributed to the review and editing of the manuscript.

**Funding** This study/project is funded by the National Institute for Health Research (NIHR) School for Primary Care Research (project reference 318).

**Competing interests** None declared.

**Patient consent for publication** Not required.

**Ethics approval** The study complies with the declaration of Helsinki, and the protocol was approved by the London - City & East Research Ethics Committee in June 2016 (ref 16/LO/1173).

**Provenance and peer review** Not commissioned; externally peer reviewed.

**Data availability statement** No data are available.

**ORCID iD**
Mark Lown http://orcid.org/0000-0001-8309-568X

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
