## [Reviewer comments · BMJ Open]

ARTICLE DETAILS

TITLE (PROVISIONAL)	Patients' views about screening for atrial fibrillation (AF): a qualitative study in primary care
AUTHORS	`Lown, Mark; Wilcox, Christopher R; hughes, Stephanie; Santer, Miriam; Lewith, George; Moore, Michael; Little, Paul

VERSION 1 – REVIEW

REVIEWER	John Mandrola Baptist Health Louisville, USA
REVIEW RETURNED	19-Aug-2019

GENERAL COMMENTS	This qualitative study addresses important issues surrounding screening for AF. Previous to this work, I had not seen any data on patients' views about AF screening. Since AF screening can begin a cascade of diagnostic and treatment, it seems obvious that we should strive to improve patients' understanding of the condition—especially since consumer devices will soon allow people to screen themselves. The paper is well written and clear. The poor understanding of AF and the potential downsides of screening programs struck me as a worthy contribution to the literature. I have a couple of inquiries. 1) It would be helpful to know what “inductive thematic analysis” is without going to a reference. 2) The authors say the sample size of 15 patients is appropriate and sufficient to achieve saturation. 15 patients, all of whom screened negative, seems like a small sample to me. Could the authors spend a sentence of two explaining why this is an adequate sample. 3) I am no expert in qualitative research, but is it possible to tell readers exactly how many patients expressed poor understanding that AF related to the heart rhythm? For instance, the authors often write “many of the participants...” I can't help but wonder: how many? Or: were there any participants, even one or two, who knew the answers?
---

REVIEWER	Nicholas Zwar Bond University, Australia
REVIEW RETURNED	20-Oct-2019

GENERAL COMMENTS	This is a timely paper exploring patients' views about screening for atrial fibrillation. As the authors state despite the increased interest in AF screening programs there is very little research on patient understanding of both the condition and the pros and cons of screening. The results are sobering - patients even those involved in a trial on AF screening who had all been sent information about the condition, had very limited understanding of both the nature of AF and the potential negatives of screening. If these findings reflect general community knowledge of AF then considerable education will be needed before informed consent for screening is possible. The main issues with the paper are:  - how the participants were recruited is not adequately described. - inadequate description of the participants. Only age and sex is provided. Further information such as age range, education level, socio-economic status and whether English is first language should be provided if available. - limitations of the study not discussed. The key ones that come to mind are that patients are from one practice only and therefore may be quite similar to each other. Given all the participants had taken part in a trial which tend to have a recruitment bias towards more educated and health literate people the range of views in the participants be different to those recruited in other ways. There may also be important differences in view in very elderly versus older people.
--

REVIEWER	Charlotte Hespe Sydney School of Medicine, The University of Notre Dame Australia,
REVIEW RETURNED	21-Oct-2019

GENERAL COMMENTS	Thank you for an interesting study and outcomes. I have a few comments to make about your article which address the some concerns I have with the paper as it stands: 1. Outcomes - I was not really sure what the outcomes of this study were actually intended to be. The authors do stated that outcomes were to explore the views and understanding of AF and AF screening in order to determine acceptability and inform future recommendations but there was a lot of time spent talking about the lack of discussion around risks of treatment of AF and how this ethically related to any screening program. This then seemed to me to be a bias against screening for AF by the authors. The bias was evident in the amount of time spent considering the lack of concern by the participants around any risks of treatment if AF was diagnosed. This seemed to be more about the views of the authors and not related to the stated outcomes. If the authors were interested in this aspect of the screening program then it would have been better to align it better with the stated outcomes. The discussion section then stated that none of the participants discussed potential harms of the screening that would arise with a positive finding. The methods section had stated that saturation of themes had occurred after the 15 interviews.
--

	2. Limitations. There were only 4 males in the study cohort and only one practice setting used for the study. There was no commentary about why this limitation was not addressed. No commentary was provided about the social setting for this practice and the limitation that this might have been - or how it might have affected / influenced the findings of this study. As an external set of eyes I felt that they should / could have broadened the cohort by conducting interviews at a different practice location and ensuring a stronger representation of men. This may have then increased the likelihood of more opinions emerging during the interviews. I was also unsure as to why there was commentary about no-one being interviewed with AF given the program was to detect AF and not about monitoring or treatment of AF. I felt that the discussion section was then long with a lot of time spent on the topic of treatment of AF rather than the actual issues of screening and what the patients did or did not want to know.
--	--

REVIEWER	Nicole Lowres University of Sydney, Australia
REVIEW RETURNED	01-Nov-2019

GENERAL COMMENTS	This study addressed an important and potentially under-researched issue in stroke prevention by examining patient attitudes toward AF screening. While there is growing momentum in international guidelines supporting opportunistic AF screening for patients aged 65 years and over, we know comparatively less about the patient experience/patient perspective of screening, and this needs to be understood if screening is to be successful for stroke reduction. The context in which the study has taken place is well described, and the results of the study are clearly outlined. There are some substantive and methodological issues that need to be addressed in the manuscript, which are detailed below.  • Please provide some more detail of the inductive thematic analysis method and triangulation methods that were employed. Please explain the rationale for using inductive thematic analysis over other analytic approaches. Additional information about the validity of the inductive thematic method used needs to be provided. • Was there some triangulation of the generation of the themes from the data? Or was triangulation only employed in the coding of the data, and was triangulation blinded by each coder? • Some detail about the aims and method of the SAFETY AF study needs to be provided to understand the context of the study. This could be briefly summarised in the Methods. • Please provide detail in the methods about how screening was undertaken in the SAFETY AF trial- did the patient self-screen, or was screening facilitated/performed by a health practitioner? • It would also be beneficial to describe and explain the AF screening devices that were used in the study- as these factors have direct impact on the patient experience of screening, and their limitations for generalising to AF screening needs to be discussed. • What are the socio-demographic characteristics of the population that (agreed to and) were interviewed? E.g education level, health literacy, employment, income, co-morbidities etc. These factors are known to affect attitudes to health care. Did these factors vary substantially from the total population that was screened? i.e. was the population of 15 people that agreed to be interviewed representative of the larger population of 418 participants?
---

	 • More than 50% (n=19/34) of people invited declined the invitation to undergo interview. What were the reasons given for declining participation? Other comments:  • Page 3- The main strengths and limitations discussed here relate to the methodological aspects of the study. It would add some strength here to highlight some of the substantive strengths and limitations. • Conclusions, page 3- please outline how decision aides are likely to be utilised, and how they can help improve patient knowledge. • One of the limitations was that the study was conducted at one GP surgery- please outline how this is likely to have contributed to the limitations- what were the unique characteristics of the practice that would have made the results more or less generalizable to other practices? • Page 2 line 39- 'screening was' rather than 'screening were' • Page 6, line 26- 'previously'
--	--

VERSION 1 – AUTHOR RESPONSE

Comment	Response	Changes
Reviewer: 1		
It would be helpful to know what "inductive thematic analysis" is without going to a reference	Thank you for this comment. We have added a description in the text.	Added text "this approach aims to generate new theory from patterns or themes of meaning within data" page 4.
The authors say the sample size of 15 patients is appropriate and sufficient to achieve saturation. 15 patients, all of whom screened negative, seems like a small sample to me. Could the authors spend a sentence of two explaining why this is an adequate sample.	Thank you for this comment. We have added more clarification to the text	Text added "the authors reviewed themes emerging during coding and were confident saturation had been reached when no new themes of interest were arising." Page 4.
is it possible to tell readers exactly how many patients expressed poor understanding that AF related to the heart rhythm? For instance, the authors often write "many of the participants..." I can't help but wonder: how many? Or: were there any participants, even one or two, who knew the answers?	Thank you for this comment we have now expanded this section and given more examples of the understanding expressed by participants.	We have added numbers to clarify (6/15) described heart rhythm abnormalities and (2/15) described stroke risk. Also added another description of the condition "it's a condition that is related to the blood circulation" to page 6.
Reviewer: 2		
how the participants were recruited is not adequately described.	Thank you for this comment, participants were approached sequentially in order of randomisation to the main trial.	We have added "Participants who consented to the qualitative study were approached sequentially (in order of randomisation to the main study) by telephone." Page 5.

inadequate description of the participants. Only age and sex is provided. Further information such as age range, education level, socio-economic status and whether English is first language should be provided if available.	Thank you for this comment. Age range is now provided. We did not measure level of education. All participants were fluent in English and socio-economic range of practice has been provided.	Age range = 65-73 (page 6). "All participants were fluent in English" added to page 6. The index of multiple deprivation score for the participating GP practice was 18.2 and the income deprivation score affecting older people (aged > 60) index was 17.2, reflecting low levels of deprivation. Added to page 6.
- limitations of the study not discussed. The key ones that come to mind are that patients are from one practice only and therefore may be quite similar to each other. Given all the participants had taken part in a trial which tend to have a recruitment bias towards more educated and health literate people the range of views in the participants be different to those recruited in other ways. There may also be important differences in view in very elderly versus older people.	Thank you for this comment. We agree and have added comments relating to use of a single GP practice and similarities.	"Participants were recruited from a single surgery and may have had similar views, not representative of the wider population." However, they were form an area of low deprivation an as they had taken part in a trial were likely to be more health literate than the wider population who may have a poorer understanding of AF. Page 3.
Reviewer: 3		
Outcomes - I was not really sure what the outcomes of this study were actually intended to be. The authors do stated that outcomes were to explore the views and understanding of AF and AF screening in order to determine acceptability and inform future recommendations but tthere was a lot of time spent talking about the lack of discussion around risks of treatment of AF and how this ethically related to any screening program. This then seemed to me to be a bias against screening for AF by the authors. The bias was evident in the amount of time spent considering the lack of concern by the participants around any risks of treatment if AF was diagnosed. This seemed to be more about the views of the authors and not related to the stated outcomes. If the authors were interested in this aspect of the screening program then it would have been better to align it better with the stated outcomes.	Thank you for this comment. In terms of the objectives of the study, we were aiming to gather data relating to the wider aspects of screening, particularly regarding the acceptability of undertaking a screening and the potential for prolonged screening. We felt that the lack of understanding of AF and the potential for harm as screening were very interesting and important findings which may need to be addressed as part of AF screening or case finding. We have attempted to redress the balance by discussing the importance placed on screening by patients before discussing the consideration of potential harms.	Added sentence "Patients expressed positive opinions towards screening, diagnosing AF at an early stage and the potential to save costs." to discussion section, page 10.

The discussion section then stated that none of the participants discussed potential harms of the screening that would arise with a positive finding.	We feel the lack of consideration of harms with screening is also well documented in the literature.	
There were only 4 males in the study cohort and only one practice setting used for the study. There was no commentary about why this limitation was not addressed. No commentary was provided about the social setting for this practice and the limitation that this might have been - or how it might have affected / influenced the findings of this study. As an external set of eyes I felt that they should / could have broadened the cohort by conducting interviews at a different practice location and ensuring a stronger representation of men. This may have then increased the likelihood of more opinions emerging during the interviews. I was also unsure as to why there was commentary about no-one being interviewed with AF given the program was to detect AF and not about monitoring or treatment of AF. I felt that the discussion section was then long with a lot of time spent on the topic of treatment of AF rather than the actual issues of screening and what the patients did or did not want to know.	Thank you for this comment. We have added comments surrounding the limited number of male participants and social demographics, the use of a single practice and the inclusion of negative screen participants. We have added additional comments in the discussion regarding participant views on screening. We have acknowledged “The study had good representation in terms of a typical screening population aged over 65 years, although there were only four male participants and all participants were from a single GP surgery in England.” and “The participants all had a negative screen which may have affected their attitudes.” In the limitations section.	Added “Participants were recruited from a single surgery and may have had similar views, not representative of the wider population. However, they were from an area of low deprivation and as they had taken part in a trial were likely to be more health literate than the wider population who may have a poorer understanding of AF. (page 3). Added “The index of multiple deprivation score for the participating GP practice was 18.2 and the income deprivation score affecting older people (> 60 years) index was 17.2, reflecting low levels of deprivation.” Page 6. Added sentence “Patients expressed positive opinions towards screening, diagnosing AF at an early stage and the potential to save costs.” to discussion section, page 10. Added sentence. “Additionally, more than half of those invited for screening responded and were willing to take part in the SAFETY trial.” Page 10.
Reviewer 4		
Please provide some more detail of the inductive thematic analysis method and triangulation methods that were employed. Please explain the rationale for	Thank you for this comment. We used inductive thematic analysis as there is little data in the literature on views on screening for AF and we did	Added “this approach aims to generate new theory from patterns or themes of meaning within data as there is little data in the literature on

using inductive thematic analysis over other analytic approaches. Additional information about the validity of the inductive thematic method used needs to be provided.	not have preconceived concepts.	patient view on AF screening." Page 5.
Was there some triangulation of the generation of the themes from the data? Or was triangulation only employed in the coding of the data, and was triangulation blinded by each coder?	Thank you for this comment. Data was coded and themes generated independently by two researchers and the main themes were reviewed and refined within the whole research team, in line with methods for carrying out thematic research described in reference 12.	We have added the sentence "Main themes and sub-themes were generated independently by ML and CW then reviewed and refined through further discussions within the team." Page 5.
Some detail about the aims and method of the SAFETY AF study needs to be provided to understand the context of the study. This could be briefly summarised in the Methods.	Thank you for this comment. We have clarified some of the detail in the methods section.	Individuals aged over 65 years both with and without a coded diagnosis of AF in their medical records were invited by their GP to a single nurse-led screening visit in order to test the accuracy of several devices (a blood pressure meter, a single-lead device and two ECG sensing consumer devices) for the detection of AF. Page4-5.
Please provide detail in the methods about how screening was undertaken in the SAFETY AF trial- did the patient self-screen, or was screening facilitated/performed by a health practitioner?	Thank you for this comment – as above.	
It would also be beneficial to describe and explain the AF screening devices that were used in the study- as these factors have direct impact on the patient experience of screening, and their limitations for generalising to AF screening needs to be discussed.	Thank you for this comment. Participants were instructed on how to use the devices and allowed to ask questions. We have discussed how the use of a BP meter may have confounded understanding of AF.	Added. "Research nurses explained how to use the devices and participants were able to ask questions about AF and the devices during the visit. " Page 5. Added "Although a blood pressure meter was employed as one of the screening devices, the other 3 devices used in the trial measured ECG signals and participants had a 12-Lead ECG and were informed that it was a definitive diagnostic test." Page 11
What are the socio-demographic characteristics of the population that (agreed to and) were interviewed? E.g education level,	Thank you for this comment. We have practice level demographic information and did not collect individual level	Added "The index of multiple deprivation score for the participating GP practice was 18.2 and the income

health literacy, employment, income, co-morbidities etc. These factors are known to affect attitudes to health care. Did these factors vary substantially from the total population that was screened? i.e. was the population of 15 people that agreed to be interviewed representative of the larger population of 418 participants?	socioeconomic / educational data. We have included this in the analysis and believe our cohort may have a better understanding than the wider population.	deprivation score affecting older people (> 60 years) index was 17.2, reflecting low levels of deprivation.” Page 6
More than 50% (n=19/34) of people invited declined the invitation to undergo interview. What were the reasons given for declining participation?	Thank you for this comment. We did not record the reasons for declining to participate in this qualitative study.	Added “We did not record reasons for declining to participate.” To page 5
Page 3- The main strengths and limitations discussed here relate to the methodological aspects of the study. It would add some strength here to highlight some of the substantive strengths and limitations.	Thank you for this comment. We have added additional information to the strengths and limitations section.	
Conclusions, page 3- please outline how decision aides are likely to be utilised, and how they can help improve patient knowledge.	Thank you for this comment. Decision aids could be used to explain the diagnostic test, conditions that could be diagnosed, quantitative information relating to diagnostic accuracy and the risks and benefits of treatment. They could also be used to promote clarification of patients’ preferences about the screening and potential consequences in order to improve patient knowledge.	We have added “Decision aids could be used to explain the diagnostic test, conditions that could be diagnosed, quantitative information relating to diagnostic accuracy and the risks and benefits of treatment. They could also be used to promote clarification of patients’ preferences about the screening and potential consequences in order to improve patient knowledge.” To page
One of the limitations was that the study was conducted at one GP surgery- please outline how this is likely to have contributed to the limitations- what were the unique characteristics of the practice that would have made the results more or less generalizable to other practices?	Thank you for this comment. The participants were from a surgery with a low level of deprivation and were likely to be more health literate than the wider population.	Added “Participants were recruited from a single surgery and may have had similar views, not representative of the wider population. However, they were from an area of low deprivation and as they had taken part in a trial were likely to be more health literate than the wider population who may have a poorer understanding of AF.” Page 3.
Page 2 line 39- ‘screening was’ rather than ‘screening were’	Thank you – corrected.	“The screening was generally...” Page 2.

Page 6, line 26- 'previously'	Thank you – corrected.	“it needs to be highlighted that all participants had received an information sheet, which described AF as...” Page 6.
------------------------	--

VERSION 2 – REVIEW

REVIEWER	john mandrola Baptist Health Louisville, USA
REVIEW RETURNED	18-Dec-2019

GENERAL COMMENTS	Given the exuberance, I would even suggest, irrational exuberance over AF screening, the authors address an important and timely topic with this research. At first, I found the sample size small, but as I read through the information obtained from the 15 interviews, I felt more comfortable. Obviously the key findings were that patients had a poor understanding of AF, and perhaps more crucially, they had little appreciation for the limits and dangers of screening. This is huge. Though this was qualitative research, I found the interviewed patients near total lack of realization of overdiagnosis, misdiagnosis and harm from therapy quite striking. These findings deserve publication. A few critiques: I am still a bit unclear on inductive thematic analysis. I would ask the authors to dumb down the explanation of what this means. For instance, the comment “the authors reviewed themes emerging during coding and were confident saturation had been reached when no new themes of interest were arising.” I am perplexed. The manuscript is way too long. The introduction could be shortened substantially. Most readers understand AF, and the great push to screen for AF. The discussion, too, could be shortened as there is much repetition about the main findings. This impact of this paper would be greater if it were less than 2500 words. The authors discuss the limitations of the study in the summary bullet points, but I feel the limitations of this small single-center study from a discrete area of the UK deserve an entire paragraph. A more robust discussion of the limitations, in my opinion, increases the impact of the paper.
---

REVIEWER	Nicholas Zwar Bond University, Australia
REVIEW RETURNED	04-Dec-2019

GENERAL COMMENTS	Good responses to reviewers' comments and an improved paper.
--

REVIEWER	Charlotte Hesse The University of Notre Dame Australia, Sydney, Australia
REVIEW RETURNED	10-Dec-2019

GENERAL COMMENTS	This revised paper has addressed most of my concerns with your first paper. I do however feel that it would be a more complete paper if you had provided a slightly fuller discussion about why this practice was chosen from the 3 participating in the full project and how the authors feel the skew of female patients may have affected the results. There is also still an absence of commentary about where this patient cohort sits within the normal social demographics of UK as a whole. Stating that they are "reflecting low levels of deprivation" does not assist me as a reader in understanding where they might sit compared to the rest of your population. Thank you for submitting a much more balanced and interesting paper.
---

REVIEWER	Nicole Lowres and Katrina Giskes Heart Research Institute, Australia
REVIEW RETURNED	05-Dec-2019

GENERAL COMMENTS	Thank you for the revisions of the manuscript. Overall, I feel that the revisions have improved the manuscript. However, I feel that some of the responses (mostly related to the methods) were addressed with less rigor than other responses. Specifically, the response to the questions raised about Inductive thematic analysis requires further justification for choosing this method and an explanation of the advantages of using this method in contrast to other qualitative analysis methods. I feel the explanation of coding could be expanded As all patients interviewed were screen negative – it is likely that some views were not captured in the interviews and subsequent analysis and this should be mentioned in the limitations As demographics other than age and gender were not collected for this study, then this is also a limitation. Perhaps reference to the demographics of the larger study could be presented. The second sentence added in Results (participant characteristics) is hard to understand for the average reader. Rewording of this sentence would make this easier to understand – perhaps along the lines of “suggesting a more socio-economically advantaged group”
---

VERSION 2 – AUTHOR RESPONSE

Comment	Response	Changes
Reviewer: 1		
The manuscript is way too long. The introduction could be shortened substantially. Most readers understand AF, and the great push to screen for AF. The discussion, too, could be shortened as there is much repetition about the main findings.	Thank you for this comment. We have now reduced the introduction to two succinct paragraphs (reducing the word count in this section from 329 to 219 words) We thank you for this comment and have eliminated repetition in the discussion section and reduced the word count from 1260 to 1010. These changes have shortened the manuscript by 400 words. We do not feel we can remove more text than this without sacrificing key content of this qualitative paper, including responses to other reviewers' comments, but would make further efforts to do so if the editor felt it essential.	Introduction paragraphs 1 & 2 shortened (page 4). Discussion & practice and policy implications section (pages 10-12)
The authors discuss the limitations of the study in the summary bullet points, but I feel the limitations of this small single-center study from a discrete area of the UK deserve an entire paragraph. A more robust discussion of the limitations, in my opinion, increases the impact of the paper	Thank you for this comment and we agree that a more detailed discussion of the limitations increases the impact.	Added paragraph "Patients who chose not to participate...." To page 11.
Reviewer: 3		
It would be a more complete paper if you had provided a slightly fuller discussion about why this practice was chosen from the 3 participating in the full project and how the authors feel the skew of female patients may have affected the results.	Thank you for this comment. We chose the first recruiting practice in order that the data could be collected earlier in the study to allow sufficient time for transcription and analysis during the study period. The other two recruiting sites which are in areas of low deprivation.	We have added the text "(from the first recruiting site of 3 which all had low levels of deprivation)." To Methods section: data Collection – page 5.
There is also still an absence of commentary about where this patient cohort sits within the normal social demographics of UK as a whole. Stating that they are "reflecting low levels of deprivation" does not assist me	Thank you for this comment – we have now commented on how this sits within the rest of England. (The index data has recently been updated.)	We have added the text "17.7 (2nd quintile (from low levels of deprivation to high) in England)..." to the results section – page 6.

as a reader in understanding where they might sit compared to the rest of your population.		
Reviewer: 4		
The questions raised about Inductive thematic analysis requires further justification for choosing this method and an explanation of the advantages of using this method in contrast to other qualitative analysis methods.	Thank you for this comment. We used inductive thematic analysis as we had little prior data or predetermined theory on AF; the actual data itself is used to derive the structure of analysis using a flexible and reflective technique rather than a more systematic approach of classifying and quantifying data. This is a widely used method of analysing qualitative data and is described fully in reference 12. We have added text but not gone into extensive detail in the interest of brevity.	We have added the text “(as we had little prior data or predetermined theory on AF; to section Methods: Data analysis – page 5.
I feel the explanation of coding could be expanded As all patients interviewed were screen negative – it is likely that some views were not captured in the interviews and subsequent analysis and this should be mentioned in the limitations As demographics other than age and gender were not collected for this study, then this is also a limitation. Perhaps reference to the demographics of the larger study could be presented.	Thank you for this comment. We have added a limitations paragraph to the main section detailing the potential for different views amongst those declining to participate, patients from other backgrounds / demographics and also screen positive patients. We have added text stating that we did not record individual demographic data. We have referenced the demographics of the full SAFETY study	We have added the paragraph, “Patients who chose not to participate in the trial may have had different views... Discussion section – page 11. Added text “although we did not record individual demographic data. To study limitations section page 3. Added text “The average age of the participants in the SAFETY trial was 73.9 years and 43% were female.” to results section – page 6.
The second sentence added in Results (participant characteristics) is hard to understand for the average reader. Rewording of this sentence would make this easier to understand – perhaps along the lines of “suggesting a more socio-economically advantaged group”	Thank you for this comment – we have now included the centiles for England.	Added “The index of multiple deprivation score for the participating GP practice was 17.7 (2nd best quintile in England) and the income deprivation score affecting older people (> 60 years) index was 17.3 (middle quintile) to results section – page 6.

VERSION 3 – REVIEW

REVIEWER	Charlotte Hesse The University of Notre Dame Australia School of Medicine, Sydney
REVIEW RETURNED	31-Jan-2020

GENERAL COMMENTS	I have had the benefit of reviewing this article previously so was very interested to see how the authors would address the issues raised in previous reviews. Previously I had felt that this article was worth publishing. I am therefore disappointed with this submission as I don't feel fundamental concerns about this paper have been addressed and I feel that there are fundamental flaws in the project that cannot be overcome by further revision. Firstly, I continue to question why there were only 15 interviews conducted with 11 being female when the majority of those screened for AF in the main study were actually male (also noting lines 12-13 page 6 that 26.7% of the participants were male with 43% of the overall trial were female, rather than 26.7 versus 53%). While I accept that the interviewer may have stopped after 15 interviews if they felt data "saturation" had been achieved I do not feel the authors address this issue and I still left asking "Why?". Secondly, the authors continue to convey a strong personal bias against screening programs. This bias is apparent in their discussion about "Practice and Policy Implications" (page 13- lines 30-49) where there are strong statements about what "should" be done prior to screening with no references with evidence to back their claims - for example " Adequate information should include the risk of bleeding with anti-coagulation treatment if AF is detected". As a practicing clinician I would dispute the "should" statements made in this paragraph and ask for the authors to provide evidence as currently they appear to be expressions of personal opinion. There is no doubt that this information "must" be given to a patient if they are found to have AF to ensure fully informed consent prior to treatment but suggest that there are ethical issues and questions that arise when stating that this "should" be provided prior to a screening program. Indeed, as a reader, the results of the interviews seem to suggest the participants were happy with the information provided to participate in the screening program and were actually also happy to be re-screened. There is no mention of concern that participants felt under informed. Interestingly, this issue may have been better addressed by participants if the team had include a cohort who had actually tested positive and who could have reported back about whether they felt they should have been better informed prior to screening. The analysis results suggest several did not understand what AF was, but this is actually different from them feeling uninformed about participating in a screening program. Looking at the questions asked by the interviewer it also appears that participants were not asked if they had read and felt that they understood the material provided prior to being asked to describe their understanding of AF.
--

REVIEWER	Nicole Lowres and Katrina Giskes Heart Research Institute, Sydney, Australia
REVIEW RETURNED	24-Jan-2020

GENERAL COMMENTS	In my opinion the authors have adequately addressed the remaining concerns of the reviewers.
--

VERSION 3 – AUTHOR RESPONSE

Comment	Response	Changes
Reviewer: 3		
Firstly, I continue to question why there were only 15 interviews conducted with 11 being female when the majority of those screened for AF in the main study were actually male (also noting lines 12-13 page 6 that 26.7% of the participants were male with 43% of the overall trial were female, rather than 26.7 versus 53%). While I accept that the interviewer may have stopped after 15 interviews if they felt data "saturation" had been achieved I do not feel the authors address this issue and I still left asking "Why?"	Thank you for this comment. We met regularly during the data collection phase to discuss the emerging data in terms of depth and quantity and agreed that saturation had been obtained. The main themes and sub-themes emerged early in the coding process. The male / female imbalance was as a result of the sequential convenience sample of the participants who consented to the qualitative sub-study. We have acknowledged this in the limitations section which has now been expanded in the main text.	We have amended a sentence in the methods section: "The study team held regular meetings during the data collection phase and we assessed the data for saturation of main themes and searched for disconfirming cases (the authors reviewed themes emerging during coding and were confident saturation had been reached when no new themes of interest were arising approximately half way through the interviews.)" The limitations section has been expanded in the main text. "The high proportion of female participants occurred due to the sequential convenience sample of the participants who consented to the qualitative sub-study and may have affected the results."
Secondly, the authors continue to convey a strong personal bias against screening programs. This bias is apparent in their discussion about "Practice and Policy Implications" (page 13- lines 30-49) where there are strong statements about what "should" be done prior to screening with no references with evidence to back their claims - for example " Adequate information should include the risk of bleeding with anti-coagulation treatment if AF is detected". As a practicing clinician I would dispute the "should" statements made in this	Thank you for this comment. We did not intend to convey a negative bias against screening and have re-worded this section accordingly. We agree with the comment and have not made any recommendations but mentioned consideration of	We have amended sentences in the practice and policy implications section (page 12-13). "Although participants in this study did not raise concerns regarding a lack of clear information, the importance of providing information of risks as well as benefits of screening is well-established in other screening programmes. For patients considering screening for AF,

paragraph and ask for the authors to provide evidence as currently they appear to be expressions of personal opinion. There is no doubt that this information "must" be given to a patient if they are found to have AF to ensure fully informed consent prior to treatment but suggest that there are ethical issues and questions that arise when stating that this "should" be provided prior to a screening program.	discussing risks as well as benefits of treatments as is the case in other screening programmes. We have also removed the last paragraph containing the statement about heterogeneity in stroke risk data as this could be interpreted as this could be interpreted as biased against screening.	healthcare professionals could consider providing information about the risk of bleeding with anticoagulation treatment if AF were detected, in addition to potential benefits. ”
Indeed, as a reader, the results of the interviews seem to suggest the participants were happy with the information provided to participate in the screening program and were actually also happy to be re-screened. There is no mention of concern that participants felt under informed.	Thank you for this comment, we have acknowledged this in the principal findings section in the discussion, and acknowledged that participants did not raise concerns about the information provided in the practice and policy implications section.	We have added “However, more than half of those invited for the screening trial were willing to take part in the SAFETY trial¹⁴ with the information we provided.” to the principal findings section, page 10. We have added “Although participants in this study did not raise concerns regarding a lack of clear information” to practice and policy implications section, page 12.
The analysis results suggest several did not understand what AF was, but this is actually different from them feeling uninformed about participating in a screening program.	Thank you for this statement and we agree and have added statements accordingly as per above.	As above. We have added “However, more than half of those invited for the screening trial were willing to take part in the SAFETY trial¹⁴ with the information we provided.” to the principal findings section, page 10. We have added “Although participants in this study did not raise concerns regarding a lack of clear information” to practice and policy implications section, page 12.
There is no mention of concern that participants felt under informed. Interestingly, this issue may have been better addressed by participants if the team had include a cohort who had actually tested positive and who could have reported back about whether they felt they should have been better informed prior to screening.	Thank you for this comment. We agree that screen positive patients may have expressed differing opinions and may have highlighted feeling uninformed about participating in a screening study.	We have added the text “We were unable to explore the views of screen positive patients who may have felt they could have been better informed prior to screening” to the strengths and limitations section in the discussion, page 11.